# Recent Advances in the Diagnosis and Management of Retrograde Ejaculation: A Narrative Review

**DOI:** 10.3390/diagnostics15060726

**Published:** 2025-03-14

**Authors:** Charalampos Konstantinidis, Athanasios Zachariou, Evangelini Evgeni, Selahittin Çayan, Luca Boeri, Ashok Agarwal

**Affiliations:** 1Urology & Neuro-Urology Unit, National Rehabilitation Center, 13122 Athens, Greece; konstantinidischaralampos@yahoo.com; 2Andrology Unit, Urology Department, Ioannina University, 45500 Ioannina, Greece; 3Cryogonia Cryopreservation Bank, 11526 Athens, Greece; lina.evgeni@cryogonia.gr; 4Department of Urology, Andrology Unit, University of Mersin School of Medicine, 32133 Mersin, Turkey; selcayan@hotmail.com; 5Department of Urology, Fondazione IRCCS Ca’ Granda—Ospedale Maggiore Policlinico, 20122 Milan, Italy; dr.lucaboeri@gmail.com; 6Cleveland Clinic Foundation, Cleveland, OH 44195, USA; agarwaa32099@outlook.com

**Keywords:** retrograde ejaculation, pathophysiology, diagnosis, treatment, sperm retrieval, post ejaculation urine sample

## Abstract

Retrograde ejaculation (RE) is a condition where the forward expulsion of seminal fluid is impaired, leading to infertility and psychological distress in affected individuals. This narrative review examines the etiology, pathophysiology, diagnosis, and management of RE, emphasizing its impact on male fertility. RE may result in the partial or complete absence of the ejaculate. Causes of RE include anatomical, neurological, pharmacological, and endocrine factors, with common triggers such as diabetes, spinal cord injury, and prostate surgery. Diagnosis primarily involves the patient history, a laboratory analysis of post-ejaculatory urine samples, and advanced imaging techniques. Management strategies for RE include pharmacological interventions, surgical approaches, and assisted reproductive technologies (ARTs). Sympathomimetic and parasympatholytic agents have demonstrated some success but are limited by side effects and variability in outcomes. ARTs, particularly with sperm retrieved from post-ejaculatory urine, offer a viable alternative for conception, with techniques such as urine alkalization and advanced sperm processing showing promising results. Despite these advancements, treatment efficacy remains inconsistent, with many studies relying on small sample sizes and lacking robust clinical trials. Future research should focus on refining diagnostic tools, optimizing ART protocols, and developing minimally invasive treatments. By addressing these gaps, healthcare providers can improve fertility outcomes and the quality of life for patients with RE.

## 1. Introduction

Retrograde ejaculation (RE) is a condition in which the forward ejaculation of seminal fluid is partially or completely absent, often due to underlying organic factors. Individuals with RE frequently report a noticeable reduction in seminal fluid volume, experience a dry ejaculate, or detect sperm in their urine following orgasm [1]. Estimating the prevalence of RE is challenging; however, studies suggest that it affects approximately 14–18% of individuals with ejaculatory disorders [2] and contributes to 0.3–2% of male infertility cases [3]. Prevalence rates of RE in infertility evaluations vary widely; for example, studies conducted in Northern Italy have reported rates ranging from 1.4% to 16% [4].

The diagnosis of RE is relatively simple and relies on a combination of patient history and laboratory testing. Patients typically report low seminal volume (partial) or a total lack of ejaculation (complete) following orgasm. Confirmation involves analyzing a post-masturbatory or post-orgasmic urine sample. Seminal plasma is often colorless, and its pH level can provide a quick diagnostic clue. The presence of 10–15 spermatozoa per high-power field or a concentration of more than 1 million sperm in the sample confirms the diagnosis. In addition to its effect on fertility, RE can cause significant psychological distress, particularly for individuals troubled by the unexpected absence of ejaculation [5].

First identified in 1814, RE was initially poorly understood, with early researchers often conflating it with anejaculation [6]. From a physiological perspective, RE results from a combination of factors, including defective bladder function and impaired sphincter control. These issues can be congenital or acquired, primary or secondary [7]. Common causes include conditions such as multiple sclerosis [8], spinal cord injuries [9], and diabetes [10], as well as the use of alpha-blocker medications [11] or complications following prostate surgery [12].

Although research has shown some success in retrieving sperm from individuals with retrograde ejaculation, there remain notable gaps in understanding related to diagnosis, recovery methods, handling procedures, and in vitro fertilization treatments [13]. This study seeks to compile and organize the existing knowledge in this domain, offering practical recommendations for effectively diagnosing and managing these cases in clinical practice. The main objective of this research was to review and synthesize current insights on RE and success rates in retrograde ejaculation cases, emphasizing the specific methods used. A secondary objective was to highlight and suggest areas for future research to address gaps and enhance understanding in this field.

## 2. Methods

A comprehensive literature review was conducted using three primary databases: Medline via PubMed, Web of Science, and Scopus. The search employed targeted keywords, including “retrograde ejaculation”, in combination with “treatment”, “fertility”, and “infertility”. To enhance the scope of the review, the reference lists of relevant articles were also meticulously examined, ensuring an in-depth exploration of the available literature.

The inclusion criteria centered on studies involving adult males experiencing RE related to fertility and infertility. Priority was given to studies that thoroughly described detailed treatment interventions. Eligible research designs included randomized controlled trials (RCTs), as well as prospective and retrospective studies, with a preference for English-language publications to maintain consistency in data interpretation. The manuscript primarily incorporates the recent literature published within the last decade (2015–2024) to ensure an up-to-date and relevant synthesis of current knowledge was performed.

To maintain a focused review, specific exclusion criteria were applied. Studies addressing ejaculation dysfunction conditions other than RE were excluded to concentrate on RE-specific infertility. Articles that did not directly discuss fertility complications were also omitted. Furthermore, duplicate studies were systematically identified and removed to ensure the reliability and accuracy of the findings.

The choice of a narrative review methodology over a systematic review was intentional to provide a comprehensive and flexible synthesis of the literature, facilitating a broader discussion of pathophysiological mechanisms, diagnostic approaches, and clinical implications. Unlike systematic reviews, which follow strict inclusion criteria and predefined protocols, a narrative review enables the integration of diverse sources, expert opinions, and evolving concepts that may not yet be extensively studied in a standardized manner.

## 3. Physiology of Ejaculation

The process of ejaculation comprises two distinct phases, namely emission and expulsion. While not formally classified as an independent occurrence, pre-ejaculation, which occurs during foreplay, encompasses the closure of the bladder neck to prevent retrograde ejaculation and the initiation of contractions in the prostate that serve to lubricate the urethra [14].

Seminal emission involves the orchestrated conveyance of seminal fluid and sperm via peristaltic movements, originating from the cauda epididymis, vas deferens, seminal vesicles, and prostate ultimately reaching the prostatic urethra. During emission, the contents of the ampullary vasa deferentia are conveyed into the prostatic urethra, where they mix with prostatic fluid. The completion of the emission phase is marked by the expulsion of the contents of the seminal vesicles into the prostatic urethra. This entire process is triggered by a synchronized sympathetic release from neurons originating in thoracolumbar spinal segments T10–L2, traversing through the hypogastric nerve and reaching the pelvic region [15,16].

In the course of ejaculation, sperm is discharged in a forward direction, succeeded by the contraction of the bladder neck to prevent retrograde semen flow into the bladder. The regulation of the expulsion phase is overseen by somatic nerves originating from spinal segments S2–S4 [17]. These nerves control the coordinated contractions of both the voluntary external urethral sphincter and the muscles of the pelvic floor [18]. It is important to distinguish RE from anejaculation, which is characterized by the lack of emission and the absence of sperm in the post-orgasmic urine sample.

## 4. Pathophysiology and Causes of Retrograde Ejaculation

RE is typically beyond an individual’s voluntary control due to its complex etiology, and a thorough diagnostic evaluation is necessary to accurately determine the underlying cause. Multiple conditions leading to bladder neck dysfunction and RE are shown in Table 1.

Bladder neck dysfunction induced by pharmacological agents is a prevalent factor contributing to RE. This may involve the effect of prescribed medications for BPH, such as uroselective alpha-blockers like silodosin, tamsulosin, and alfuzosin. Silodosin is an alpha-blocker with high uroselectivity and exerts the most significant impact on ejaculation [19]. Research suggests that alpha-blockers reduce the contraction capacity of the seminal vesicles, leading to a lack of emission or anejaculation instead of RE [20].

Additionally, medications for mood disorders like selective serotonin reuptake inhibitors (SSRIs), including but not limited to fluoxetine and sertraline, and antipsychotics like chlorpromazine, thioridazine, and risperidone, are recognized as potential contributors to RE [21].

The majority of men undergoing transurethral surgery for benign prostatic hyperplasia (BPH) using classical techniques experience permanent RE. From an embryological perspective, the prostate and ejaculatory ducts originate from distinct structures. The mesonephric duct gives rise to the ejaculatory ducts, while the prostate develops from the primitive urogenital sinus [22]. The ejaculatory ducts lack muscular cells and rely on the contractions of the seminal vesicles, vas deferens, and prostatic smooth muscle during the emission phase. Their primary function is transportation rather than production or contraction. Anatomical studies of the ejaculatory ducts are critical in understanding retrograde ejaculation mechanisms, particularly following surgical interventions for BPH.

To date, only one study, conducted by Malalasekera et al., has specifically examined the anatomical course and relationships of the ejaculatory ducts, using six cadaveric subjects over 50 years old [23]. This study introduced the concept of the peri-montanal zone, located 7.5 mm laterally and 10 mm proximally from the verumontanum. Across varying prostate sizes, it was found that in 95% of cases, the ejaculatory ducts traverse this zone.

Because they keep the internal urethral sphincter intact, the newer minimally invasive techniques for BPH management have lower rates of RE [24,25]. The current tools for assessing RE associated with BPH treatment have significant limitations. Optimized evaluation should include detailed assessments of both baseline and post-treatment ejaculatory function, addressing specific aspects such as force, volume, sensation, and the degree of distress, rather than relying on generalized questions measured on a broad scale [26].

Bladder neck disorders encompass both congenital broad bladder necks and surgically induced injuries to the bladder neck, which can result in insufficient closure and retrograde ejaculation. Additionally, urethral conditions such as urethral valve syndrome or strictures caused by trauma or inflammation can heighten urethral resistance. This increased resistance may obstruct semen flow during ejaculation, ultimately leading to retrograde ejaculation [27]. Pelvic radiation therapy for prostate cancer treatment produces a lack of ejaculation in about 89% of patients [28].

Nerve damage linked to uncontrolled diabetes mellitus, multiple sclerosis, retroperitoneal lymph node dissection without nerve sparing, or spinal cord injury can result in functional impairment in the closure of the bladder neck [29]. In diabetic men, ejaculatory dysfunction represents a noteworthy sexual consequence, affecting approximately 35–50% of individuals with diabetes mellitus [10]. For testicular cancer patients undergoing retroperitoneal lymph node dissection, fertility issues arising from RE pose a significant concern, particularly for younger participants with ejaculatory dysfunction who had not yet started or considered starting a family at the time of diagnosis. Despite the recurrent identification of ejaculatory dysfunction in patients with retroperitoneal lymph node dissection across various studies, effective treatments remain limited [30].

Men with spinal cord injuries (SCIs) may experience retrograde ejaculation at a higher rate, ranging from 8% to 37%. During ejaculation, the closure of the bladder neck is typically regulated by the sympathetic nervous system, which originates from the T12–L2 spinal cord segments. In males with SCIs, disruptions to these sympathetic pathways, such as those caused by nerve damage or lesions, can prevent proper bladder neck closure, resulting in semen being redirected into the bladder instead of being expelled through the urethra [9].

Endocrine factors, such as hypothyroidism, can manifest as mononeuropathy or sensorimotor polyneuropathy. If these conditions affect the nerves responsible for controlling ejaculation, they may result in retrograde ejaculation [5]. Similarly, hyperprolactinemia can disrupt dopamine metabolism, which is essential for the function of adrenergic receptor agonists [31]. This disruption can impair the proper closure of the bladder neck, leading to retrograde ejaculation [32].

## 5. Diagnosis of Retrograde Ejaculation

Diagnosing RE is generally straightforward and involves evaluating the patient’s medical history along with laboratory tests. Patients often report either a reduced semen volume (partial RE) or a complete absence of ejaculation (complete RE) after orgasm. The occurrence of aspermia or hypospermia accompanied by oligozoospermia should prompt the consideration of RE [33]. Additionally, seminal plasma is typically clear in appearance, and its pH level can serve as a useful diagnostic indicator.

The detection and analysis of sperm in post-ejaculatory urine samples (PEUSs) may assist in diagnosing RE, though it is not a definitive indicator on its own [34]. The presence of 10–15 spermatozoa per high-power field or a concentration of more than 1 million sperm in the sample confirms the diagnosis. Additionally, measuring fructose levels in PEUSs using the indol method, which generates a yellow-orange colorimetric reaction, can provide further confirmation of RE [35].

In healthy adult males, normal urinary fructose levels typically range from undetectable to 0.5298 mmol/L, even after high fructose intake, as determined by advanced techniques like ultra-performance liquid chromatography–mass spectrometry (UPLC-MS/MS) [36,37]. In contrast, semen contains significantly higher fructose concentrations—approximately 25 times greater than urine—with averages of 14.2 mmol/L in men with proven fertility [38]. An increase in fructose levels in PEUSs beyond baseline, resulting from the mixing of semen with urine, serves as a reliable indicator of RE. It is important to highlight that fructose occurs naturally in various fruits and vegetables. Additionally, it is commonly used as an added sweetener in processed foods, beverages, honey, and syrups.

The presence of sperm in PEUSs has been observed in 60–70% of men with confirmed paternity, casting doubt on its reliability as the sole diagnostic criterion for RE [39]. Mehta et al. noted that in these cases, sperm is predominantly found in the initial fraction of urine. This observation has led to the hypothesis that sperm detected in PEUSs may often originate from retained semen in the urethra rather than indicating true RE [40]. Infertile men more prevalently present sperm in post-ejaculation urine analysis, especially in cases of decreased volume of ejaculation and anejaculation, where RE is usually confirmed after PEUS analysis [41].

Severe oligozoospermia with low semen volume has been strongly correlated with increased sperm counts in PEUSs, with an inverse relation between the number of retrieved spermatozoa and the ejaculated seminal volume [42]. In a study by Mieusset et al., an attempt was made to define limits for the evaluation of the expected chance of RE being present in men with repeated low semen volumes by use of the retro-ejaculatory index (R-ratio). This indicator evaluates the total number of sperm recovered in PEUSs as a percentage of the total number of sperm found in both semen and PEUSs [41]. The authors suggested that the evaluation of sperm counts in PEUSs can be systematically recommended as an add-on to a second semen analysis following the initial one showing low semen volume, in order to confirm its constant presence, followed by a further evaluation of partial RE. An R-value higher than the range of 7.1 ± 8.3% may be considered as indicative of the need for further investigation for RE or the absence/obstruction in the seminal vesicles, either isolated or in combination with an absence of the vas deferens. However, this suggestion must be further validated by additional prospective multicenter studies [43].

Advanced methods have been introduced to enhance diagnostic accuracy. Harini and Vickram utilized MATLAB-based theoretical models to compare sperm flow patterns, as well as maximum and mean velocity, between retrograde ejaculation and normospermia. An independent samples *t*-test revealed no statistically significant differences in flow rate, maximum velocity, or mean velocity between the retrograde ejaculation and non-retrograde groups (*p* = 0.193, 0.374, 0.067). However, mathematical simulations suggested that normospermia exhibited higher values in flow rate (0.09817 m^3^/s), mean velocity (0.03125 m^2^/s), and maximum velocity (0.0625 m^3^/s) compared to retrograde ejaculation [44].

Suprapubic bladder aspiration is one of the suggested methods and is performed after orgasm [33]. The presence of sperm in both PEUSs and bladder urine, with higher concentrations in PEUSs, is indicative of true RE. In contrast, sperm detected exclusively in PEUSs suggests retained semen in the urethra [13].

Another effective diagnostic method involves the real-time monitoring of ejaculation using transrectal ultrasound, which facilitate the visualization of the bladder neck’s behavior during emission and expulsion, providing a reliable means of determining whether it remains open (Figure 1) [45,46].

## 6. Treatment Options for Retrograde Ejaculation

The pharmacological intervention should ideally be individualized according to the causes, symptoms, and erectile dysfunction. Before starting treatment, it is essential to rule out other potentially reversible causes, including medications linked to the occurrence of RE (Table 2). Treatment is particularly critical for individuals with RE who are planning to have children.

Options for managing RE, both pharmacological and surgical, are limited [47]. The existing literature primarily consists of small case series and randomized trials, highlighting the need for additional research, particularly randomized placebo-controlled studies [3].

When reversible causes such as anatomical abnormalities or diabetes are not present, pharmacological treatment is typically the primary approach. This strategy aims to increase sympathetic tone in the internal urethral sphincter and the vas deferens, thereby preventing semen from flowing backward into the bladder. The treatment involves either stimulating the sympathetic nervous system or inhibiting the parasympathetic nervous system [30].

Sympathomimetic medications are particularly beneficial for patients with conditions such as diabetic neuropathy or for those with a loss of emission due to retroperitoneal sympathetic nerve damage post-surgery. Frequently used drugs in this category include synephrine, pseudoephedrine hydrochloride, ephedrine, phenylpropanolamine, and midodrine. These medications can augment seminal output and restore natural conception in some aspermic patients [48,49,50]. However, side effects such as dizziness, restlessness, dry mouth, nausea, sweating, tachycardia, and hypertension are common. These drugs should be prescribed cautiously, especially for diabetic patients with cardiovascular risk [17,51,52].

Parasympatholytic medications, such as brompheniramine maleate and imipramine, demonstrate a success rate of 22% when used alone, which increases to 39% when combined with sympathomimetics [2]. While combination therapy appears more effective, the small sample sizes in studies make statistical validation challenging. Other approaches include buspirone [53] and transurethral injections of collagen or Deflux^®^ (a gel composed of dextranomer microspheres and hyaluronic acid), as a method to create resistance against the backward flow of semen. However, these studies are generally limited by small sample sizes, and the long-term impacts on lower urinary tract function and bladder emptying remain unclear [54,55].

Kurbatov et al.’s study represents a particularly related patient cohort for collagen treatment. In their research, 24 men with type 1 diabetes mellitus and RE unresponsive to imipramine were randomized to receive either endourethral collagen injections or saline injections in a double-blinded manner. At a 12-month follow-up, the group treated with collagen demonstrated substantial enhancements in ejaculatory volume and all sperm parameters compared to the saline group. However, no long-term follow-up data for these patients have been published [56].

Surgical treatments have shown some success, but reports are limited to a few small case series from the 1980s [2]. In cases where there is surgical damage of the bladder neck due to BPH management, reconstruction of the bladder neck has been described in small case series with promising results [57]. As a result, other alternatives, such as sperm retrieval from post-ejaculatory urine or the testes for assisted reproductive technologies (ART), should be explored before considering surgical interventions [58].

## 7. How to Deal with Post-Ejaculation Urine Samples?

Sperm retrieval in patients with RE is mainly based on efficiently processing urine samples collected post-ejaculation. This approach is valid but also challenging, as the urine environment, with its hostile pH and osmolality conditions, is detrimental to sperm motility and viability.

### 7.1. Patient Preparation in the Pre-Collection Period

To maximize viable sperm retrieval from post-ejaculatory urine samples (PEUSs), specific patient instructions must be provided. Both semen and PEUSs should be collected in the Andrology laboratory. Samples collected at home and transferred to the laboratory must be rejected from analysis.

If sperm retrieval is intended for use in ARTs, the patient is commonly advised to consume fluids that contain alkaline substances, e.g., sodium bicarbonate, acetazolamide, and potassium citrate, trying to neutralize the highly acidic pH of the urine fluid to which the spermatozoa will be exposed after ejaculation [59].

### 7.2. Laboratory Preparation

When a patient with RE is expected in the Andrology laboratory, specific preparations are warranted. Two separate sterile collection containers are explicitly marked and sperm culture media are warmed at 37 °C for 20 min. The collection process is initiated by partially emptying the patient’s bladder by urination. Then, ejaculation is attempted in a sterile container marked as ‘SEMEN’ with complete orgasm, even if no seminal fluid is secreted (anejaculation). Immediately afterward, the patient provides a urine sample in another sterile container labeled ‘URINE’, which contains 9 mL of the pre-heated sperm wash medium [59,60].

### 7.3. Laboratory Processing of Post-Ejaculation Urine Samples

The urine sample is transferred into numerous 15 mL conical sterile tubes and centrifuged at 300× *g* for 10 min. The supernatant is discarded from all tubes. The pellet in the first tube is reconstituted in 1–2 mL of warm wash medium and transferred to the second one. This process continues until all reconstituted pellets are transferred to the last tube and diluted with the wash medium to a final volume of 20 mL. The initial urine volume as well as the final suspension volume is recorded.

An aliquot of well-mixed resuspended sample is then used for assessment. Sperm concentration, motility, vitality, and morphology are evaluated according to the 6th edition of the WHO manual for semen analysis and processing [59]. It is important to note that the examined sample has been derived from a semen–culture medium suspension. The collected semen fluid (if present) is also evaluated for the basic seminal parameters [60]. In the case of anejaculation, the complete absence of ejaculated fluid is recorded. In case the semen sample is highly viscous, an additional step of treatment with 5 mg of chymotrypsin and incubation for 10 min at 37 °C is performed before semen analysis [59].

Sperm processing for use in ARTs is performed according to the protocol described by Agarwal et al. using a sterile double-density gradient by careful layering of the reconstituted specimen on the upper phase of the gradient [59]. A post-wash semen analysis is also performed and sperm recovery is calculated. The resulting pellet is suspended in a culture medium containing human serum albumin in Earle’s/Hank’s and phosphate-buffered medium. The derived sperm mixture is then used in the ART method. This process has been reported to result in a pregnancy rate of 15% per cycle, according to 15 studies applying either IUI or IVF/intracytoplasmic sperm injection (ICSI) [59].

### 7.4. Methods of Optimization of Sperm Viability in Post-Ejaculation Urine Samples

The most challenging factor in the attempt of sperm retrieval from PEUSs is the optimization of the high osmolality and low pH in urine, compared to semen (normally 300–380 mOsm/kg and 7.2–8.2, respectively). The ingestion of oral alkalizing agents during patient preparation may increase sperm motility from 42.4% to 99.1% in prepared PEUSs. A 15% pregnancy rate per cycle has been reported in ART cycles using retrieved sperm in cases of intrauterine insemination (IUI) or in vitro fertilization (IVF)/intracytoplasmic sperm injection (ICSI) [42,59].

The Hotchkiss method of emptying the bladder by a catheter followed by washing and instillation of a small quantity of Ringer’s lactate has also been applied successfully with a reported 25% pregnancy rate and a 28% live birth rate in IVF and ICSI. A modified version of this method has been reported to produce similar results of live birth rates per transfer (28%) in ICSI using frozen retrieved sperm in RE patients with low correspondence to medical treatment [42,59].

The method of ejaculation with a full bladder into a sterile container with Baker’s buffer medium has been reported to have limited efficiency in IUI, mainly due to the lack of standardization in the available studies [42,59].

If these methods fail to yield viable sperm from PEUSs, surgical options such as micro testicular sperm extraction (mTESE) may be considered. It is generally advised to begin with the least invasive techniques and progress to more complex and invasive methods only if simpler approaches are unsuccessful.

## 8. Conclusions

Retrograde ejaculation (RE) is a multifaceted condition with significant implications for male fertility and psychological well-being. Despite its relatively low prevalence among infertile male cases, RE represents a unique challenge due to its varied etiologies and the intricacies of its diagnosis and management.

Treatment options for RE are diverse, ranging from pharmacological interventions, such as sympathomimetic and parasympatholytic agents, to surgical approaches and ARTs. Pharmacological treatments can restore antegrade ejaculation in some cases, but are limited by variable efficacy and potential side effects. Surgical interventions, while effective in certain contexts, are often invasive and carry associated risks. ARTs, particularly involving sperm retrieval from post-ejaculatory urine, have emerged as viable non-surgical alternatives, offering a pathway to conception for couples affected by RE. Techniques such as urine alkalization, the Hotchkiss method, and advanced sperm processing protocols have demonstrated success, albeit with varying outcomes.

Despite these advancements, notable gaps in knowledge persist. The current literature is heavily reliant on small case series and lacks robust randomized controlled trials to establish standardized treatment protocols. Moreover, the optimization of sperm retrieval methods and ART outcomes requires further investigation to enhance success rates and address the unique challenges posed by RE. The development of minimally invasive diagnostic tools and targeted therapeutic strategies remains a critical area for future research.

In conclusion, while significant progress has been made in understanding and managing retrograde ejaculation, ongoing research and innovation are essential to address its complexities. By bridging current knowledge gaps and refining treatment approaches, healthcare providers can better support patients with RE, improving both fertility outcomes and overall quality of life.

## Figures and Tables

**Figure 1 diagnostics-15-00726-f001:**
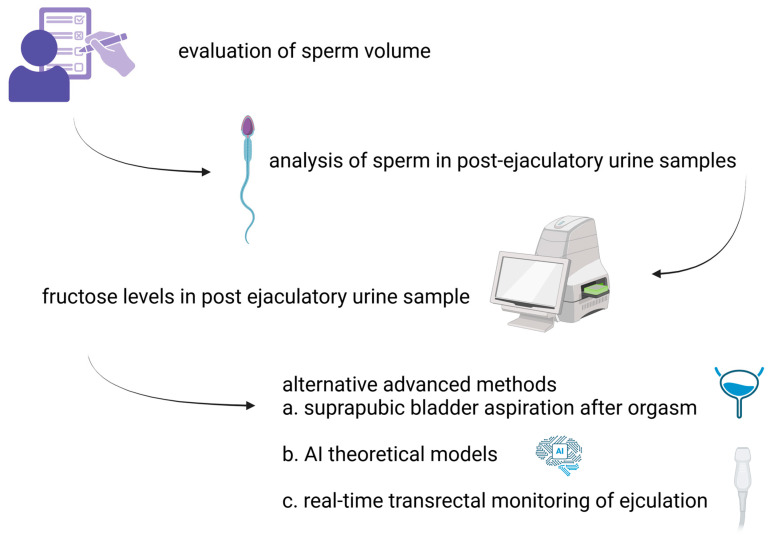
Step-by-step approach to diagnosing RE.

**Table 1 diagnostics-15-00726-t001:** Causes of retrograde ejaculation (RE).

Pharmacological treatments	Uroselective alpha-1 adrenoceptor treatment for BPH
	Antidepressants
	Antipsychotics
Bladder neck—anatomical causes	Transurethral procedures for benign prostatic hyperplasia (BPH)
	Surgical interventions for bladder outlet obstruction
	Exposure to radiation therapy
Neurological causes	Neuropathy from diabetes mellitus
	Spinal cord trauma
	Cauda equina syndrome
	Multiple sclerosis
	Retroperitoneal lymph node dissection without nerve preservation
	Sympathectomy procedures
Endocrine factors	Hypothyroidism
	Hyperprolactinemia

**Table 2 diagnostics-15-00726-t002:** Medical treatment of retrograde ejaculation.

Treatment Agents	Suggested Treatment Dose
Ephedrine	60–120 mg/daily
Pseudoephedrine	60–120 mg/daily
Brompheniramine	8–16 mg/daily
Imipramine	50–100 mg/daily
Amoxapine	50 mg/daily

## Data Availability

Not applicable.

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
