# Peer review of "Recent Advances in the Diagnosis and Management of Retrograde Ejaculation: A Narrative Review"

_diagnostics, 2025, doi:10.3390/diagnostics15060726_

Round 1
Reviewer 1 Report
Comments and Suggestions for Authors
This manuscript provides a comprehensive narrative review of retrograde ejaculation (RE), covering its etiology, pathophysiology, diagnosis, and management with a strong emphasis on its impact on male fertility. The authors discuss the various causes of RE, including anatomical, neurological, pharmacological, and endocrine factors, as well as common triggers such as diabetes, spinal cord injury, and prostate surgery. Diagnostic strategies rely on patient history, laboratory analysis of post-ejaculatory urine, and imaging techniques, while management options include pharmacological treatments, surgical interventions, and assisted reproductive technologies (ART). The review acknowledges that treatment efficacy remains inconsistent, with a lack of robust clinical trials and small sample sizes limiting the available evidence. Future research directions include improving diagnostic tools, optimizing ART protocols, and developing minimally invasive therapies.
General comments
- The manuscript is clear, well-organized, and thoroughly researched. It provides an up-to-date synthesis of the literature on RE and highlights relevant clinical implications. The writing is precise, and the content is well-structured, making it accessible to both clinicians and researchers in the field of andrology and reproductive medicine. The review successfully integrates pathophysiological insights with current diagnostic and treatment strategies, making it a valuable contribution to the literature.
Minor comments
- The introduction is too long and could be more concise while still providing essential background information. The section includes historical details, epidemiology, pathophysiology, and future research suggestions, some of which could be moved to separate sections or streamlined.
- Consider reorganizing the manuscript following the SANRA guidelines for structured narrative reviews.
- Consider to add a flowchart illustrating the step-by-step approach to diagnosing RE (including history-taking, post-ejaculatory urine analysis, fructose testing, and imaging techniques) would make the diagnostic process easier to follow.
- While the search strategy is well described, it would be useful to explicitly mention the timeframe of the literature search to clarify how recent the included studies are.
- Consider briefly discussing why narrative review methodology was chosen over systematic review methodology.
Author Response
REVIEWER 1
1. The introduction is too long and could be more concise while still providing essential background information. The section includes historical details, epidemiology, pathophysiology, and future research suggestions, some of which could be moved to separate sections or streamlined.
Answer: Thank you for your thoughtful comments, we deleted lines 59-63, 66-69.
2. Consider reorganizing the manuscript following the SANRA guidelines for structured narrative reviews.
Answer: Thank you for your thoughtful comments and for recognizing the manuscript’s clarity, organization, and relevance in your General Comments. I appreciate your suggestion regarding the SANRA guidelines. Given your positive assessment of the manuscript’s structure and coherence, I believe that the current organization effectively presents the synthesis of the literature while ensuring accessibility for both clinicians and researchers.
- Consider to add a flowchart illustrating the step-by-step approach to diagnosing RE (including history-taking, post-ejaculatory urine analysis, fructose testing, and imaging techniques) would make the diagnostic process easier to follow.
Answer: Thank you very much for your insightful comment. We added figure 1 according to your suggestions.
3. While the search strategy is well described, it would be useful to explicitly mention the timeframe of the literature search to clarify how recent the included studies are.
Answer: Thank you very much for your insightful comment, we added lines 87-89
4. Consider briefly discussing why narrative review methodology was chosen over systematic review methodology.
Answer: Thank you for your insightful comment. We have added lines 95-101 to clarify our preference for a narrative review over a systematic review.
Reviewer 2 Report
Comments and Suggestions for Authors
In their manuscript „Recent Advances in the Diagnosis and Management of Retrograde Ejaculation: A Narrative Review”, Konstantinidis and colleagues summarize the current knowledge on the pathophysiology, diagnosis and therapy of retrograde ejaculation. The authors present a comprehensive summary of the available literature, and conclude that - despite progress in this field has occurred - there are still aspects in retrograde ejaculation (RE) warranting further research. Strengths of the manuscript is the usage of mainly current literature < 10 years old and the clinical relevant topic. Weaknesses are in the discussion of the different diagnostic and treatment options and the chimeric structure of the manuscript with parts being a narrative review and parts similar to a methods paper. I would recommend publication after some revisions.
Here are my concerns in detail:
1) Throughout the manuscript: orthography and semantics are excellent. However, there are some redundancies, especially regarding the Abstract and the main text (l. 14 vs. l. 36) and some strange phrases (l. 17: RE may result in partially of… instead of ..occur…; l. 27: showing promising outcomes). I would recommend to fix these issues in the production process.
2) l. 117: this sentence is puzzling, as the authors present several etiologies in the following paragraphs. I would recommend rephrasing it.
3) Regarding chapter 4: I would recommend describing the different causes of RE also in relation to the frequency of occurrence.
4) Regarding the diagnosis: l. 239 - 242: the supporting reference for the ultrasound-guided monitoring of ejaculation is 20 years old. Are there more current references for this technique?
5) Regarding the treatment options (chapter 6): the references given in this chapter are rather old. This fact is in contrary to the title of the manuscript indication recent advances in the management of RE. It is not comprehensible, to which extent the named options currently play a role in the therapy of RE.
6) Regarding chapter 7: this chapter is in structure and aim different from the rest of the manuscript. While Chapter 1-6 give an overview in the style of a (narrative) review, chapter 7 is structured like a protocol in the style of a method paper. Furthermore, there are no references in this chapter, despite the book chapter of the last author and the WHO manual. I would recommend adjusting this chapter in structure and content to the rest of the manuscript and induce some more (recent) reviews.
Finally, I want to thank the authors for sharing their review with the clinical community. Best regards.
Author Response
REVIEWER 2
1) Throughout the manuscript: orthography and semantics are excellent. However, there are some redundancies, especially regarding the Abstract and the main text (l. 14 vs. l. 36) and some strange phrases (l. 17: RE may result in partially of… instead of ..occur…; l. 27: showing promising outcomes). I would recommend to fix these issues in the production process.
Answer: Thank you for your valuable comments, we made all the alterations according to your suggestions.
2) l. 117: this sentence is puzzling, as the authors present several etiologies in the following paragraphs. I would recommend rephrasing it.
Answer: thank you very much for your comment. We rewrote the sentence. “RE is typically beyond an individual’s voluntary control due to its complex etiology. A thorough diagnostic evaluation is necessary to accurately determine the underlying cause.”
3) Regarding chapter 4: I would recommend describing the different causes of RE also in relation to the frequency of occurrence.
Answer: Thank you for your valuable comment and made the proper alterations.
4) Regarding the diagnosis: l. 239 - 242: the supporting reference for the ultrasound-guided monitoring of ejaculation is 20 years old. Are there more current references for this technique?
Answer: Thank you for your valuable comment. We added a more recent citation, according to your suggestion – reference 48.
Hara R, Nagai A, Fujii T, et al. Practical application of color Doppler ultrasonography in patients with ejaculatory dysfunction. Int J Urol 2015;22:609-611.
5) Regarding the treatment options (chapter 6): the references given in this chapter are rather old. This fact is in contrary to the title of the manuscript indication recent advances in the management of RE. It is not comprehensible, to which extent the named options currently play a role in the therapy of RE.
Answer: Thank you for your insightful comment, we have updated five references.
6) Regarding chapter 7: this chapter is in structure and aim different from the rest of the manuscript. While Chapter 1-6 give an overview in the style of a (narrative) review, chapter 7 is structured like a protocol in the style of a method paper. Furthermore, there are no references in this chapter, despite the book chapter of the last author and the WHO manual. I would recommend adjusting this chapter in structure and content to the rest of the manuscript and induce some more (recent) reviews.
Answer: Thank you for your insightful feedback. Chapter 7 is presented in a structured, protocol-like format because it describes a laboratory process, which requires a clear, step-by-step methodology. Unlike the narrative review style of Chapters 1–6, this chapter focuses on practical implementation rather than literature synthesis. Furthermore, we added in references the few papers describing the procedures for sperm retrieval in RE.